# Current Understanding of Asthma Pathogenesis and Biomarkers

**DOI:** 10.3390/cells11172764

**Published:** 2022-09-05

**Authors:** Nazia Habib, Muhammad Asghar Pasha, Dale D. Tang

**Affiliations:** 1Department of Molecular and Cellular Physiology, Albany Medical College, Albany, NY 12208, USA; 2Department of Allergy, Asthma, and Immunology, Albany Medical College, Albany, NY 12208, USA

**Keywords:** asthma, smooth muscle, cytokine, inflammation, biomarker

## Abstract

Asthma is a heterogeneous lung disease with variable phenotypes (clinical presentations) and distinctive endotypes (mechanisms). Over the last decade, considerable efforts have been made to dissect the cellular and molecular mechanisms of asthma. Aberrant T helper type 2 (Th2) inflammation is the most important pathological process for asthma, which is mediated by Th2 cytokines, such as interleukin (IL)-5, IL-4, and IL-13. Approximately 50% of mild-to-moderate asthma and a large portion of severe asthma is induced by Th2-dependent inflammation. Th2-low asthma can be mediated by non-Th2 cytokines, including IL-17 and tumor necrosis factor-α. There is emerging evidence to demonstrate that inflammation-independent processes also contribute to asthma pathogenesis. Protein kinases, adapter protein, microRNAs, ORMDL3, and gasdermin B are newly identified molecules that drive asthma progression, independent of inflammation. Eosinophils, IgE, fractional exhaled nitric oxide, and periostin are practical biomarkers for Th2-high asthma. Sputum neutrophils are easily used to diagnose Th2-low asthma. Despite progress, more studies are needed to delineate complex endotypes of asthma and to identify new and practical biomarkers for better diagnosis, classification, and treatment.

Asthma is a heterogeneous lung disease that affects more than 300 million people worldwide [1]. Asthma is characterized by variable airflow obstruction and airway hyperresponsiveness (AHR), leading to episodic and reversible bronchoconstriction, because of an exaggerated airway-narrowing response to many environmental triggers, such as allergens. Traditionally, the illness is classified into two groups: extrinsic and intrinsic asthma. Extrinsic asthma is also known as allergic asthma, which is caused by allergens and mainly attributed to aberrant T helper type 2 (Th2) inflammation. Intrinsic asthma is triggered by various factors, such as aspirin, pulmonary infection, exercise, cold, stress, obesity, etc.

Recently, based on the status of Th2 inflammation, the disease has been classified into two groups: Th2-high and Th2-low asthma. Th2-high asthma is characterized by eosinophilic airway inflammation, which is associated with increased blood eosinophil counts or elevations of fractional exhaled nitric oxide (FeNo), whereas Th2-low asthma includes neutrophilic asthma and paucigranulocytic asthma. The coexistence of eosinophilic and neutrophilic airway inflammation is considered mixed granulocytic asthma [1,2]. The pathological mechanisms of asthma are complex, varying in different phenotypes caused by different environmental triggers, ages, obesity, genetic factors, etc. In addition to airway inflammation, there is emerging evidence to suggest that inflammation-independent processes also contribute to asthma pathogenesis. Furthermore, biomarkers of a disease are traceable substances that are useful for diagnosis, classification, and treatment. This review is focused on the pathogenesis and biomarkers of asthma induced by allergens, infection, and pollutants.

## 1. Pathological Mechanisms of Asthma

Although asthma is classified into Th2-high and Th2-low asthma, the disease can be induced by mixed airway inflammation. Patients may have Th2-high asthma in the early stage and have Th2-low asthma in a later stage or vice versa; or Th2-high asthma and Th2-low asthma occur concurrently. Because of the complexity of asthma, we discuss the mechanisms of Th2-high asthma, Th2-low asthma, and other mechanisms separately.

### 1.1. Mechanisms of Th2-High Asthma

Th2 cells are a distinct lineage of CD4^+^ effector T cells that secrete interleukin (IL)-4, IL-5, IL-13, and IL-9. Approximately 50% of mild-to-moderate asthma and a large portion of severe asthma is induced by Th2-dependent inflammation [1,2]. Since Th2-high asthma has been reviewed in detail elsewhere [2,3,4], we summarize the key points for the mechanisms of Th-2 high asthma.

Th2 inflammation has two major phases: 1. Sensitization: When allergens enter the low airways, antigen-presenting cells process and present the allergens to Th2 cells, which secret Th2 cytokines, including IL-5, IL-4, and IL-13. IL-4 and IL-13 activate B cells, which produce IgE and bind to FcεRI of mast cells. 2. Challenge: When the same allergens enter the airways, they bind to IgE, which induces mast cells to release mediators, such as leukotrienes (LTs), histamine, and ILs. In addition, allergens act on cholinergic nerves to release acetylcholine. These mediators and neurotransmitters irritate airway smooth muscle and induce bronchoconstriction [1,2,3]. In addition, IL-5 facilitates eosinophil production, maturation, and recruitment to the lungs [5]. Eosinophils also release mediators, including major basic protein (MBP), which stimulates mast cells to release histamines and LTs. MBP also inhibits M_2_ receptor and promotes acetylcholine release from cholinergic nerves and induces bronchospasm [6]. Furthermore, IL-13 directly sensitizes airway smooth muscle contraction, stimulates epithelial cells to secret mucins, and induces fibrosis [7] (Figure 1).

Recent studies demonstrated that the airway epithelium produces cytokines in response to injury, infection, and pollutants. These epithelial-derived cytokines include thymic stromal lymphopoietin (TSLP), IL-25, and IL-33. TSLP, IL-25, and IL-33 activate type 2 innate lymphoid cells (ILC2), which generate Th2 cytokines, such as IL-5 and IL-13 and induce Th2 lung inflammation [1,2]. Additionally, there is evidence to suggest that IL-33 may directly affect mast cell activation, airway smooth muscle migration, and asthma phenotype [8] (Figure 1).

Th9 cells and IL-9 are also involved in Th2 lung inflammation [9]. Th9 cells produce the cytokines IL-9, IL-10, and IL-21; however, IL-9 is likely to contribute to asthma pathology. Because of its pleiotropic effects, IL-9 influences a variety of distinct cell types, such as T cells, B cells, mast cells, and macrophages. IL-9 may promote Th2 inflammation by activating Th2 cells and by increasing mast cell accumulation [9]. IL-9 may also activate Arg1^+^ interstitial macrophages, which secrete the chemokine CCL5. CCL5 then recruits eosinophils, T cells, and monocytes into the lungs to propagate type 2 inflammation [10] (Figure 1).

Natural killer T (NKT) cells are a distinct subset of lymphocytes that are abundant in the lungs as well as lymphoid organs. It was proposed that NKT cells secrete IL-4 and IL-13 or facilitate Th2 cells to increase production of IL-4 and IL-13 [11]. However, other studies do not support this notion [12,13].

Regulatory T cells (Tregs) are a specific CD4^+^ T cell population that act to suppress immune response, thereby maintaining homeostasis and self-tolerance. Tregs have been classified based on the expression of the transcription factor FOXP3. Tregs may inhibit asthma pathogenesis by suppressing the activation/functions of ILC2, mast cells, antigen-presenting cells, Th1/Th2/Th17 cells, eosinophils, neutrophils, and B cells [14].

One of the targets of Th2 cytokines is periostin, a matricellular protein that is a dynamically expressed non-structural protein present in the extracellular matrix. Periostin expression is upregulated by IL-4 and IL-13 in cultured bronchial epithelial cells and bronchial fibroblasts [15] and is one of the most differentially expressed bronchial epithelial genes between asthmatic patients and healthy control subjects [16]. The role of periostin in asthma is still under investigation. There are reports to suggest that periostin supports adhesion and migration of IL-5-stimulated human eosinophils and Th2 inflammation in asthma [17]. On the other hand, other studies suggest that periostin plays a protective role, rather than detrimental role in asthma. Periostin positively regulates TGF-β production, which promotes T-regulatory cell differentiation. Differentiated T cells inhibit airway inflammation and IgE production [18].

### 1.2. Mechanisms of Th2-Low Asthma

#### 1.2.1. IL-17

IL-17 has been proposed to play an important role in Th2-low asthma [19,20,21]. Variants in the IL-17 pathway genes may be related to asthma pathology [22,23]. Higher levels of IL-17 are found in serum, sputum, and bronchoalveolar lavage fluid (BALF) of patients with asthma, which is associated with asthma severity [19,20]. There are several cell types secreting IL-17 cytokines. CD4^+^ Th17 cells are one of the major sources of IL-17. Other cellular sources include major histocompatibility complex class I-restricted CD8^+^ T-cells, Natural killer T cells, mucosal-associated invariant T (MAIT) cells, ILC3 cells, and B-cells [24].

The role of IL-17 cytokines in asthma is still under investigation. IL-17 cytokines may stimulate epithelial cells and fibroblasts to release neutrophil chemoattractants CXCL1/5/8 and granulocyte–macrophage colony-stimulating factor, which recruit neutrophils to the lungs. Furthermore, IL-17A, but not IL-17F, enhances airway smooth muscle contraction [21], migration [25], and proliferation [26], which facilitates airway hyperresponsiveness (AHR) and airway remodeling, key characteristics of asthma. However, it has been proposed that IL-17 cytokines are important for maintaining the integrity of the epithelium and IL-17 cytokines may play a protective role against asthma [24] (Figure 2).

#### 1.2.2. Other Cytokines

It is known that Th1 cells secrete IL-2, interferon-γ (IFN-γ), and lymphotoxin-α and stimulate Th1 immunity, which is characterized by prominent phagocytic activity. However, recent studies suggest that some Th1 cytokines may contribute to asthma pathogenesis. Tumor necrosis factor-α (TNF-α) is a pleiotropic Th1 cytokine, which plays a role in the pathogenesis of inflammatory diseases, including allergy. Sputum TNF-α is elevated in neutrophilic and severe asthma [27]. TNF-α is proposed to synergize with IL-17 cytokines to promote neutrophil recruitment [1,24]. However, TNF-α may also promote the production of Th2 cytokines, such as IL-4, IL-5, and IL-13 [28]. Furthermore, TNF-α enhances airway smooth muscle contraction, which may contribute to the development of AHR [29] (Figure 2).

IFN-γ, IL-1β, and TNF-α have been shown to upregulate the expression of CD38 (cluster of differentiation 38), also known as cyclic ADP ribose hydrolase in airway smooth muscle cells, which may upregulate intracellular Ca^2+^ signaling and induce AHR. Knockout (KO) of CD38 reduced AHR in a murine model of asthma [30,31]. In addition, IFN-γ promotes neutrophil recruitment in the presence of IL-17 cytokines [1,24].

### 1.3. Emerging Mechanisms of Asthma

Asthma has long been viewed as an inflammatory disease. However, there is accumulating evidence to suggest that inflammation-independent processes are also associated with asthma progression. For instance, recent studies demonstrate that protein kinases, adapter proteins, and other molecules contribute to asthma pathogenesis [32,33,34,35,36,37,38,39,40] (Figure 3).

#### 1.3.1. Proteins Kinases

c-Abl (Abelson tyrosine kinase, Abl, ABL1) is a non-receptor tyrosine kinase that participates in the regulation of smooth muscle contraction, migration, and proliferation [38,41,42,43,44,45]. c-Abl is upregulated in asthmatic human airway smooth muscle (HASM) cells, which is regulated by epigenetic factors [46,47]. c-Abl KO or inhibition reduces asthma-like phenotypes in animal models of asthma [38,45]. Furthermore, c-Abl KO or inhibition diminishes Th2 cytokines in experimental asthma [38,45]. These results suggest that c-Abl is a Th2-regulatory protein rather than a Th2-dependent protein. Intriguingly, treatment with the c-Ab/KIT inhibitor imatinib relieves the symptoms of severe refractory asthma [48].

Polo-like protein kinase 1 (Plk1) is a serine/threonine kinase that plays a role in modulating smooth muscle contraction [37,49], proliferation [50,51], migration [50], mitosis [52,53], and apoptosis [40]. In asthmatic HASM cells, downregulation of miR509 leads to elevated Plk1 [50]. Smooth muscle conditional KO of Plk1 inhibits asthma progression in a murine model of asthma [52]. Plk1 may contribute to airway remodeling via promoting ASM proliferation/migration and inhibiting apoptosis [40,50,52,54]. However, Plk1 does not affect Th2 inflammation in experimental asthma [52].

p21-activated kinase (PAK) regulates smooth muscle contraction by modulating the vimentin network and paxillin complexes [54,55]. Furthermore, a PAK inhibitor or PAK KO protects mice from AHR and airway smooth muscle hyperactivity in vitro [56]. However, it is unclear whether PAK expression and activity are altered in the lungs or serum of asthmatics. Another protein kinase glycogen synthase kinase-3β (GSK-3β) is also linked to asthma pathology. Airway smooth muscle hyperplasia and hypertrophy correlate with GSK-3β phosphorylation in a mouse model of asthma [57]. GSK3 negatively regulates smooth muscle gene expression and hypertrophy. Phosphorylation of GSK3 disinhibits smooth muscle gene expression and promotes ASM hypertrophy and hyperplasia [57,58].

#### 1.3.2. Adapter Protein

Abi1 (Abelson interactor 1) is an adapter protein that regulates cell migration [59,60], smooth muscle contraction [61], and cell proliferation [39]. The human Abi1 gene is localized in the Chromosome 10p21 region. Genome-wide association studies (GWAS) suggest that Chromosome 10p21 is adjacent to a susceptible locus for asthma and related traits [62,63]. Abi1 is upregulated in asthmatic HASM cells/tissues [39]. Loss-of-function studies suggest that Abi1 contributes to aberrant HASM cell proliferation and asthma phenotype in a murine model of asthma [39].

#### 1.3.3. MicroRNAs (miRNAs)

miRNAs are evolutionarily conserved, 18–25 nucleotides, noncoding RNA molecules that control gene expression by binding to complementary sequences in the 3′ untranslated regions (3′ UTR) of target mRNAs, which degrade target mRNA and/or repress translation [64]. The levels of miR-203 are downregulated in human asthmatic ASM cells, which disinhibits c-Abl expression and promotes asthma development [46,65]. Moreover, the expression of miR-509 is lower in human asthmatic ASM cells, which is responsible for the upregulation of Plk1 and asthma progression [47,50]. miR-25 expression is associated with alterations in ASM cell phenotype, an important process for airway remodeling [66]. miR-144–3p has been shown to be associated with severe corticosteroid-dependent asthma [67].

#### 1.3.4. Others

ORMDL3 and gasdermin B. GWAS suggest that chromosome 17q21 is linked to asthma [68,69]. Chromosome 17q21 contains a cluster of genes, including ORMDL3 and gasdermin B (GSDMB) [69]. ORMDL3 may contribute to asthma progression by modulating store-operated calcium entry and lymphocyte activation [70], eosinophil trafficking and activation [71], and sphingolipid homeostasis [72]. Gasdermin B may promote AHR and airway remodeling, without affecting airway inflammation via remodeling-associated gene expression [73].

Matrix Metalloproteinases (MMPs) are calcium-dependent zinc-containing endopeptidases with more than 20 isoforms. MMPs have been linked to asthma, which is isoform dependent [74,75]. Single-nucleotide polymorphisms (SNPs) in the gene encoding MMP-12 is associated with FEV1 in children and adults with severe asthma [76]. The SNPs in the MMP-12 promoter region increase MMP-12 expression, which may activate macrophages and promote asthma progression [74]. In addition, mast cell tryptase proteolytically activates pro-MMP-1 generated by ASM, which subsequently degrade the extracellular matrix and promote ASM cell growth and airway remodeling [77]. However, MMP-2 appears to have a protective role in asthma. Mice overexpressing human MMP-2 showed a significant reduction in AHR, Th2 cytokines, and IgE compared to their wild-type counterparts [75].

## 2. Biomarkers of Asthma

As mentioned above, biomarkers of a disease are traceable substances that are useful for diagnosis, classification, and treatment. Although the omics technologies (e.g., epigenomics, genomics, transcriptomics, proteomics, metabolomics, lipidomics, etc.) and microbiome have been proposed to serve as biomarkers for asthma [78], they are still in the early stage of research. In this review, we focus on clinically practical biomarkers collected from induced sputum, blood, exhaled gases, and bronchoscopic samples.

### 2.1. Th2-High-Related Biomarkers

#### 2.1.1. Sputum Eosinophils

Eosinophils in induced sputum provide important information on asthma phenotyping and understanding of asthma pathophysiology [79]. Increased sputum eosinophil levels (>3%) have been associated with high airway inflammation, frequent asthma exacerbation, and poor asthma control [80,81].

#### 2.1.2. Blood Total Eosinophil Count (TEC)

TEC has also been considered as a non-invasive biomarker for eosinophilic inflammation [79,82,83,84]. The usage of blood eosinophil counts as a diagnostic biomarker for airway eosinophilia has been evaluated by assessing the relationship between blood and sputum eosinophil counts [85,86,87,88]. TEC increases ≥0.30 × 10^9^/L when Th2 lung inflammation and asthma exacerbations transpire. If a blood count is <0.15 × 10^9^/L, sputum eosinophilia may not be found, especially when FeNO is low (<25 ppb) [89]. However, higher TEC is also seen in patients with atopic dermatitis and other allergic diseases. Thus, the demonstration of eosinophilia is not a specific marker of Th2 lower airway inflammation. These caveats prompt physicians to use FeNO measurement, which is associated with airway inflammation [90].

#### 2.1.3. Serum IgE

Serum IgE is an immunoglobulin, which induces type 1 hypersensitivity reactions and anaphylaxis. As described earlier, IgE also plays a key role in the pathogenesis of allergic asthma. Elevated levels of IgE are correlated with patients with asthma [91]. There is an association between IgE levels, skin testing, and lung function in asthmatics. Clinical studies show that asthmatics have an inverse relationship between IgE and FEV1/FVC ratio [92]. Various clinical trials have used IgE as a biomarker to identify Th2-high asthma. Omalizumab, a recombinant human anti-IgE antibody that binds to circulating IgE at the IgE receptor binding site, blocks the activation of the mast cells and basophils. A large phase III study that recruited over 500 patients with asthma found that IgE levels are from 30 to 700 IU/mL. Omalizumab treatment was able to reduce exacerbation rates and improve quality-of-life scores [93]. However, a Cochrane review published in 2014 on the use of omalizumab questions whether there is a clear threshold level of IgE for optimal efficacy. The authors note a wide spread in the mean serum IgE levels of patients included in clinical trials, ranging from 141.5 to 508.1 IU/mL [94].

#### 2.1.4. Nitric Oxide

Nitric oxide is produced by airway epithelial cells as a result of IL-13-induced upregulation of nitric oxide synthase in the airway epithelium and is, therefore, a more specific marker of Th2 airway inflammation [95,96,97]. FeNO is a reproducible, easily measurable biomarker, indicative of AHR and a good predictor of inhaled corticosteroid (ICS) response [98,99,100]. FENO values between 25 ppb and 50 ppb (20–35 ppb in children) should be interpreted cautiously and with reference to clinical context. FENO greater than 50 ppb (>35 ppb in children) can be used to indicate that eosinophilic inflammation and, in symptomatic patients, responsiveness to corticosteroids are likely. However, FeNO may be affected by several confounders, including demographics, smoking, diet, nasal polyps, and atopic status [99,101,102,103,104]. Although most patients with raised FeNO respond to corticosteroids, some patients are resistant to corticosteroid treatment. Their FeNO is not suppressed and they have high Th2 cytokines and chemokines in sputum [90]. That said, FeNO level is a useful indication for Th2-high asthma and helps to use appropriate doses of inhaled ICS [105].

#### 2.1.5. Periostin

Periostin is upregulated by recombinant IL-4 and IL-13 in cultured bronchial epithelial cells and bronchial fibroblasts [15,16,106]. Periostin is proposed as a surrogate marker of Th2 inflammation. Serum periostin levels are significantly higher in asthmatic patients with eosinophilic airway inflammation. A logistic regression model, including sex, age, IgE levels, blood eosinophil numbers, body mass index, FeNo levels, and serum periostin levels, in 59 patients with severe asthma, showed that the serum periostin level was the best predictor of airway eosinophilia [107].

#### 2.1.6. Cytokines

Levels of IL-4, IL-5, and IL-13 in sputum and BALF are higher in asthmatics. TSLP, IL-33, and IL-25 in epithelium are elevated in asthmatic patients [106]. These cytokines are the gold standard to verify Th2-high asthma for clinical research. However, it may not be feasible for routine practice because of high costs.

These Th2-high biomarkers are being used to choose adequate biologic therapy and monitor the patients’ response to asthma treatment. For instance, higher levels in FeNO, blood eosinophils, and serum periostin (Th2-high asthma) are indications for use of the IgE antibody Omalizumab. Omalizumab treatment reduces asthma exacerbation rates and improves quality of life for this group of patients [93]. Lebrikizumab is an IgG4 humanized monoclonal antibody that specifically binds to IL-13 and blocks its function. Lebrikizumab administration was able to improve lung function. Patients with higher pretreatment levels of serum periostin had greater improvement in lung function with lebrikizumab [108]. Despite ICS therapy and an additional controller, some patients still had uncontrolled asthma. Lebrikizumab administration reduced exacerbation rate by 60% compared with a placebo in periostin-high patients and by 5% in periostin-low patients. However, lebrikizumab administration did not lead to clinically meaningful placebo-corrected improvements in asthma symptoms or quality of life [109].

### 2.2. Th2-Low-Related Biomarkers

#### 2.2.1. Sputum Neutrophils

Th2-low asthma includes late-onset asthma in middle-aged females, obesity-associated asthma, smoking-associated asthma, infection-associated asthma, and ozone-associated asthma [110,111]. Another common feature seen in Th2-low asthma is poor response to inhaled and oral corticosteroids [112,113]. Using induced sputum coupled with cytology, patients with Th2-low asthma are classed as paucigranulocytic and neutrophilic. In healthy subjects, neutrophils and macrophages are the major leukocytes in the induced sputum (median neutrophil percentage 37%). Cigarette smoking, ozone, infection, and endotoxin all increase sputum neutrophil counts. In asthma patients, sputum neutrophil count increased to 40–76% [111].

#### 2.2.2. IL-17

As described earlier, IL-17 promotes neutrophilic inflammation in asthmatics. IL-17 levels in induced sputum, BALF, and bronchial biopsies have been found to be increased in severe asthma [19,20]. Due to technical challenge and costs, measurement of sputum IL-17 has not been widely used to characterize asthma phenotype.

#### 2.2.3. Other Potential Biomarkers

TNF-α and IFN-γ contribute to the progression of Th2-low asthma [1]. IL-6 and C-reactive protein have been linked to severe asthma [111]. More studies are required to assess whether these potential biomarkers are practical in clinical settings.

### 2.3. Biomarkers Indicative of Airway Remodeling

#### 2.3.1. Bronchoscopy

Airway remodeling is characterized by airway smooth muscle thickening, epithelial metaplasia, mucus hypersecretion, and basement membrane fibrosis with deposition of abnormal extracellular matrix [2,34,39,114,115]. Remodeling is seen in adults with chronic asthma and in childhood asthma as a result of chronic airway inflammation [114,116,117]. Considerable efforts have been made to identify potential biomarkers for structural changes in asthmatics; however, there is limited success. Bronchial biopsies are the gold standard to assess remodeling but are considered an invasive procedure. A study performed morphometric analysis on bronchial biopsy specimens before and after anti-IgE (Omalizumab) treatment to investigate changes in airway remodeling after 12 months of treatment [115]. This study showed reduced reticular basement thickening in some patients. Gal-3 is a regulatory molecule acting at various stages from acute to chronic inflammation and tissue fibrogenesis. Gal-3 can be considered a reliable biomarker to predict the extent of airway remodeling in severe asthma patients treated with omalizumab. In this study, Gal-3 was the most stable biomarker associated with the prediction of airway remodeling [118]. Additionally, because Gal-3 is a matrix protein, it is feasible to detect it in serum or urine [119].

#### 2.3.2. YKL-40

YKL-40 is a chitinase-like protein that is associated with airway remodeling. In a study, YKL-40 levels in serum were increased in children with severe and therapy-resistant asthma compared to healthy children. Furthermore, serum levels of YKL-40 significantly correlate with bronchial wall thickness measured by high-resolution computerized tomography [120].

### 2.4. Genetic Risk for Asthma Development and Treatment

GWAS have implicated genetic variants in developing asthma. In particular, childhood asthma is associated with the 17q21 locus alleles. Polymorphisms of 17q21 are associated with an increased risk of exacerbations in children with asthma, despite ICS use. Single-nucleotide polymorphism (SNP) rs7216389 frequency was higher in East Asians, African Americans, and Hispanics, compared to patients of European ancestry [121]. In addition, the *ORMDL3* gene is located at the 17q21 region and plays an important role in asthma pathogenesis. Asthmatic patients have higher levels of human lung ORMDL3 and *ORMDL3* gene SNP rs8076131 [122]. IL-1receptor-like 1 (ST2) promotes asthma development by mediating the response to IL-33. *ST2* SNPs rs13431828, rs1420101, rs1921622, and rs10204137 were related to lower efficacy of ICS in children and adolescents [123].

In addition to genetic risk, many environmental factors are also important risks for asthma, although most experts do not consider environmental risks to be “biomarkers’ for asthma. Allergens (e.g., house dust mite, pollen), pollutants, bacteria, viruses, and fungi are well-known environmental risks for asthma [124,125,126]. Exposure to different environmental factors may affect different mechanisms and asthma progression. For example, IL-17A is a potential mediator to link Candida albicans sensitization and poor outcomes for asthma [127].

## 3. Clinical Differences in Th2-High and Th2-Low Asthma

### 3.1. Phenotypes of Th2-High Asthma

Phenotypes of Th2-high asthma are classified into three groups: early-onset allergic asthma, late-onset eosinophilic asthma, and aspirin-exacerbated respiratory disease (AERD) [128].

#### 3.1.1. Early Onset or “Extrinsic” Allergic Asthma 

Early onset or “extrinsic” allergic asthma is the prototype of the asthma phenotype. The clinical presentation of child-onset allergic asthma ranges from mild to severe and it is unknown whether severe asthma is the result of evolution from a milder form or instead arises de novo as a severe type during childhood. This phenotype is different from Th2-high nonatopic asthma in terms of positive allergy skin tests and increased serum-specific IgE [129].

#### 3.1.2. Late-Onset Eosinophilic Asthma

Late-onset eosinophilic asthma is a subgroup of Th2-high asthmatics with adult-onset disease, which has a distinct steroid-resistant eosinophilic phenotype of unknown molecular mechanism [130]. ICS therapy does not ameliorate airway Th2 inflammation in approximately half of this subgroup of asthmatics. Typically, these patients are older and have more severe asthma with persistent airflow obstruction [131]. The majority of these patients have comorbid chronic rhinosinusitis with nasal polyps, which generally precede asthma development. This phenotype is generally characterized by prominent blood and sputum eosinophilia, refractory to inhaled/oral corticosteroid treatment. Some of these patients have sputum neutrophilia in addition to eosinophilia, implicating Th2/Th17 inflammation [132]. This phenotype generally also has high FeNO and normal or elevated serum total IgE.

#### 3.1.3. AERD

AERD is a subset of the late-onset phenotype, characterized by asthma, chronic rhinosinusitis with nasal polyps, and cyclooxygenase (COX)-1 inhibitor-induced respiratory reactions [128]. The mechanisms of this phenotype involve dysregulated arachidonic acid (AA) and leukotriene (LT) production. COX1/2 utilizes AA to synthesize PGE_2_, which is anti-inflammatory. In contrast, 5-lipooxygenase (5-LO) uses AA to synthesize LTs, which induce airway spam. Aspirin and other nonsteroidal anti-inflammatory drugs inhibit COX1/2, which shifts the balance to the 5-LO pathway and generates more LTs [128].

### 3.2. Phenotypes of Th2-Low Asthma

Based on clinical characteristics, Th2-low asthma phenotypes have been classified into obesity-associated asthma, smoking-associated asthma, and very-late-onset asthma [128].

#### 3.2.1. Obesity-Associated Asthma

In general, obesity-associated asthma occurs in non-atopic and middle-aged women with severe symptoms, despite a moderately preserved lung function. This phenotype is not associated with eosinophilic lung inflammation. Obesity switches CD4 cells toward Th1 differentiation, which is associated with steroid refractory asthma [133]. Additionally, Th17 pathways, ILC3 that expresses both IL-17 and IL-22, and IL-6 have been associated with obesity-related asthma [128,134]. Consequently, IL-17, IL-22, and IL-6, rather than Th2 cytokines, may be clinically relevant in obese patients with severe asthma.

#### 3.2.2. Smoking-Associated Asthma

The mechanisms underlying this phenotype involve oxidative stress, which induces epigenetic modifications and causes neutrophil and macrophage activation [135]. Smoking also enhances the risk of sensitization to allergens and increases total IgE. Recently, patients with smoking history and consequent airflow obstruction but also having overlapping features of asthma (bronchodilator reversibility, eosinophilia, and atopy) have been described as having “Asthma-COPD overlap syndrome (ACOS)”. The most recently published consensus of ACOS included six criteria, three of which are major (persistent airflow limitation, tobacco smoking, and previous asthma or reversibility > 400 mL FEV1) and three minor (history of atopy or rhinitis, significant bronchodilator reversibility, and peripheral eosinophilia). Although all COPD patients have not responded to the new biologic agents, the ACOS subset may actually benefit.

#### 3.2.3. Very-Late-Onset Asthma

The age cutoff for the diagnosis of late-onset asthma is usually defined as >50–65 years [136,137]. The aging lung is associated with the loss of elastic recoil and immunosenescence, which may lead to decreased lung function. While mechanisms have not been fully understood, some studies suggest that older asthmatics have increased sputum neutrophilia, secondary to Th1 and Th17 inflammation [138,139].

## 4. Asthma-Associated Comorbidities

Asthma is often associated with a variety of comorbidities. Common reported asthma comorbidities include rhinitis, gastroesophageal reflux disease, nasal polyps, obstructive sleep apnea, hormonal disorders, vocal cord dysfunction, obesity, and psychopathologies [140,141,142]. These conditions may complicate the diagnosis and management of asthma or just coexist with asthma without obvious influence on this disease. These comorbidities could share a common pathophysiological mechanism with asthma or have different pathological processes. Future studies are required to understand how these comorbidities may interact with asthma.

## 5. Conclusions

Asthma is a heterogeneous lung disease with variable phenotypes and distinctive endotypes. In Th2-high asthma, IL-4 and IL-13 activate B cells, which produce IgE and sensitize mast cells. IL-5 promotes eosinophil recruitment to the lungs. In Th2-low asthma, IL-17 and TNF-α promote the recruitment of neutrophils to the lungs. Protein kinases, adapter protein, miRs, ORMDL3, and gasdermin B are newly identified molecules that contribute to asthma pathogenesis, independent of inflammation. Eosinophils, IgE, FeNO, and periostin are practical biomarkers for Th2-high asthma, whereas neutrophils are easily used for Th2-low asthma. Because asthma is a heterogeneous disease, more studies are required to identify new endotypes and new biomarkers to better diagnose and treat the illness.

## Figures and Tables

**Figure 1 cells-11-02764-f001:**
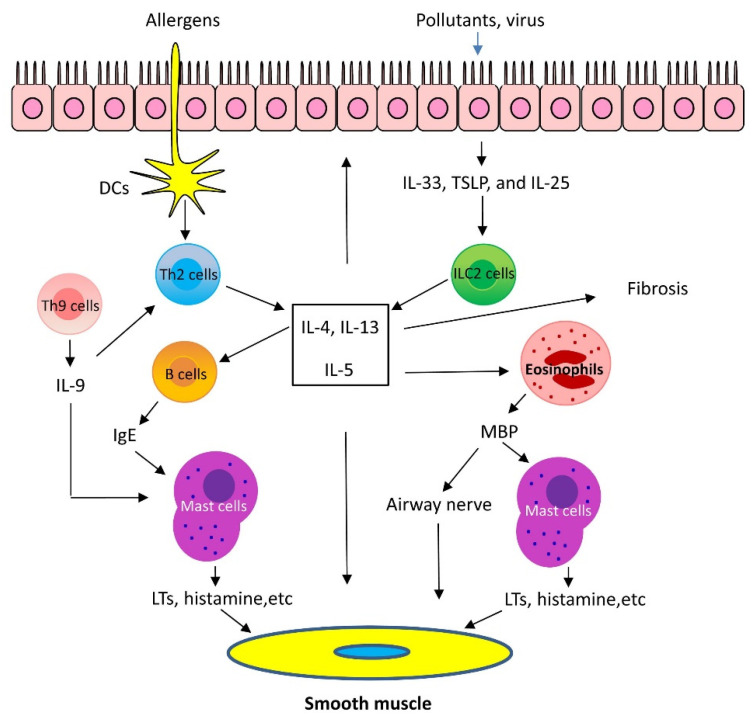
Mechanism of Th2-high asthma. When allergens enter the low airways, dendritic cells (DCs) present the allergens to Th2 cells, which secrete Th2 cytokines, including interleukin (IL)-5, IL-4, and IL-13. IL-4 and IL-13 activate B cells, which produce IgE. IgE subsequently binds to surface of mast cells. When the same allergens enter the airways, they interact with IgE, which induces mast cells to release mediators, such as leukotrienes (LTs), histamine, and ILs. These mediators irritate airway smooth muscle and induce bronchoconstriction. In addition, IL-5 facilitates eosinophil recruitment to the lungs. Eosinophils also release mediators, including major basic protein (MBP), which stimulates mast cells to release histamines and LTs. MBP also inhibits M_2_ receptor and promotes acetylcholine release from cholinergic nerves and induces bronchospasm. Furthermore, IL-13 directly sensitizes airway smooth muscle contraction, stimulates epithelial cells to secret mucins, and induces fibrosis. Th9 cells can secrete IL-9, which activates Th2 cells and promotes mast cell accumulation. Lastly, epithelium injury by infection and pollutants induces release of cytokines, including thymic stromal lymphopoietin (TSLP), IL-25, and IL-33, which activate type 2 innate lymphoid cells (ILC2) and produce Th2 cytokines, such as IL-5 and IL-13.

**Figure 2 cells-11-02764-f002:**
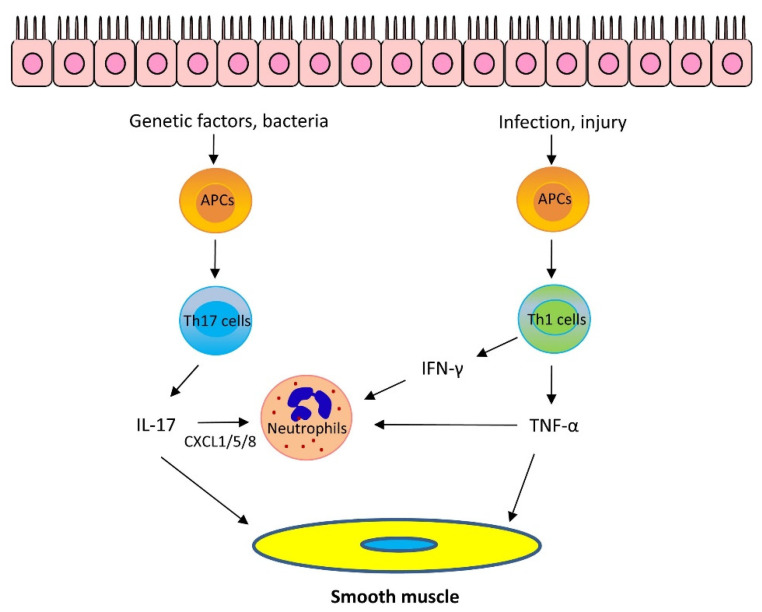
Mechanism of Th2-low asthma. Th17 cytokines: Bacteria promote Th17 cell differentiation via antigen-presenting cells (APCs). Variants in the IL-17 pathway genes also contribute to IL-17 upregulation. IL-17 can stimulate epithelial cells and fibroblasts to release neutrophil chemoattractants CXCL1/5/8 which recruit neutrophils to the lungs. Furthermore, IL-17A enhances airway smooth muscle contraction, migration, and proliferation, which facilitates AHR and airway remodeling, Th1 cytokines: Infection and epithelial injury promote Th1 cell maturation and secrete Th1 cytokines, including TNF-α and IFN-γ. TNF-α synergizes with IL-17 cytokines to promote neutrophil recruitment. Furthermore, TNF-α enhances airway smooth muscle contraction. IFN-γ and TNF-α upregulate Ca^2+^ signaling in airway smooth muscle and induces AHR. In addition, IFN-γ promotes neutrophil recruitment in the presence of IL-17 cytokines.

**Figure 3 cells-11-02764-f003:**
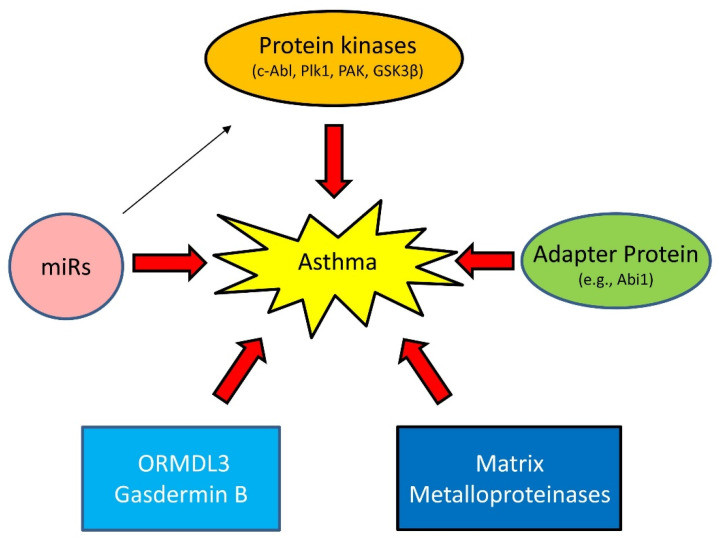
Emerging mechanisms of asthma. Asthma has long been viewed as an inflammatory disease. However, there is accumulating evidence that inflammation-independent processes also contribute to asthma progression. Genetic variance and epigenetics (e.g., miRs) affect expression of proteins, including kinases, adapter protein, ORMDL3, Gasdermin B, and matrix metalloproteinases in lung tissues, which drive asthma progression.

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
