# Peer review of "Current Understanding of Asthma Pathogenesis and Biomarkers"

_cells, 2022, doi:10.3390/cells11172764_

Round 1

Reviewer 1 Report

This is a nice review of the inflammatory mechanisms of asthma and divided up by Th2 high and low.

Unfortunately, these are not two separate situations and the inflammation cascade can include both.

Comorbidities also divide up asthma and as such, these factors could also be included, even if there is no data for such. Eg GERD, nasal polyps

Also, for a practitioner, some clinical differences beyond treatment options might be useful

Author Response

We thank the reviewer for the constructive criticisms and suggestions. We have revised the manuscript to address these concerns and suggestions.  Below are detailed our responses:

Comment 1: “This is a nice review of the inflammatory mechanisms of asthma and divided up by Th2 high and low.”

Response 1: Thanks for the comment.

Comment 2: “Unfortunately, these are not two separate situations and the inflammation cascade can include both.”

Response 2: The reviewer is correct. Although asthma is classified into Th2-high and Th2-low asthma, the disease can be induced by mixed airway inflammation. Patients may have Th2-high asthma in the early stage and have Th2-low asthma in the later stage, or vice versa; or Th2-high asthma and Th2-low asthma occur concurrently. We have revised the manuscript.

Comment 3: “Comorbidities also divide up asthma and as such, these factors could also be included, even if there is no data for such. Eg GERD, nasal polyps.”

Response 3: This is an excellent idea. We have briefly summarized comorbidities in the review.

Comment 4: “Also, for a practitioner, some clinical differences beyond treatment options might be useful”

Response 4: The reviewer is correct. We have added a subtitle for this topic. 

Reviewer 2 Report

This manuscript discusses asthma pathogenesis and biomarkers, and there are some interesting and novel viewpoints. However, considering several concerns I have below, this article may need revisions before considering published.

1.     The pathogenesis is heterogeneous which associated with various T cell subtypes, including Th2, Th17, Th9, Treg, ILC-2, and NKT cells. Th2-high and Th2-low mechanism should discuss these cells and their crosstalk to more clarify the pathogenesis in asthma.

2.     IL-33, thymic stromal lymphopoietin (TSLP), IL-25, as well as IL9, may contribute to asthma pathogenesis instead of IFN-γ or TNF-α. Author should mentioned these mechanism in the manuscript.

3.     Whether intervening on selective pathways can prevent or reverse airway wall remodelling and influence asthma's natural history is the key question. This manuscript discussed the pathogenesis and biomarker in asthma. In my suggestion, to discuss the biomarker for monitoring treatment response in asthma, especially biological agent, is important.

4.     Author discussed genetic risk for asthma development and treatment in the manuscript. However, environmental factor such as air pollution or fugal-sensitization is also important risk for asthma. For example, IL-17A is a potential mediator to link Candida albicans sensitization and poor outcomes for asthma. To discuss the environmental factors affect disease progression of asthma is crucial.

Author Response

We thank the reviewer for the constructive criticisms and suggestions. We have revised the manuscript to address these concerns and suggestions.  Below are detailed our responses:

Comment 1: “The pathogenesis is heterogeneous which associated with various T cell subtypes, including Th2, Th17, Th9, Treg, ILC-2, and NKT cells. Th2-high and Th2-low mechanism should discuss these cells and their crosstalk to more clarify the pathogenesis in asthma.”

Response 1: The reviewer is correct. We have added additional information on these mechanisms in the manuscript, particularly Th9, Tregs and NKT cells.  

Comment 2: “IL-33, thymic stromal lymphopoietin (TSLP), IL-25, as well as IL9, may contribute to asthma pathogenesis instead of IFN-γ or TNF-α. Author should mentioned these mechanism in the manuscript.”

Response: Thanks for the input. We have included these cytokines in the review. IFN-γ or TNF-α are unique because they regulate airway smooth muscle functions, which is a focus of this special issue.  

Comment 3: “Whether intervening on selective pathways can prevent or reverse airway wall remodelling and influence asthma's natural history is the key question. This manuscript discussed the pathogenesis and biomarker in asthma. In my suggestion, to discuss the biomarker for monitoring treatment response in asthma, especially biological agent, is important.”

Response 3: Correct. Biomarkers of a disease are traceable substances that are useful for diagnosis, classification, and treatment. We have discussed biomarker of monitoring asthma treatment in this review.

Comment 4: “Author discussed genetic risk for asthma development and treatment in the manuscript. However, environmental factor such as air pollution or fugal-sensitization is also important risk for asthma. For example, IL-17A is a potential mediator to link Candida albicans sensitization and poor outcomes for asthma. To discuss the environmental factors affect disease progression of asthma is crucial.”

Response 4: Correct. This review is focused on pathogenesis and biomarkers of asthma. Genetic risk can be used as biomarkers for potential asthma onset. Agree with the reviewer, environmental factors are also risk for asthma. Most experts do not think environment factors to be “biomarkers” for asthma. However, we add a sentence to briefly discuss environmental risk for asthma.  

Round 2

Reviewer 2 Report

The version of this manuscript is accepted for publication including all changes made.